# High frequency DNA rearrangement at *qγ27* creates a novel allele for Quality Protein Maize breeding

Hongjun Liu [1,6], Yongcai Huang [2,3,6], Xiaohan Li [1,6], Haihai Wang [2], Yahui Ding [1], Congbin Kang [1], Mingfei Sun [1], Fangyuan Li [1], Jiechen Wang [2], Yiting Deng[2], Xuerong Yang [1], Xing Huang[2,3], Xiaoyan Gao [2], Lingling Yuan [4], Dong An [5], Wenqin Wang [5], David R. Holding [4] & Yongrui Wu [2]*

Copy number variation (CNV) is a major source of genetic variation and often contributes to phenotypic variation in maize. The duplication at the *27-kDa γ-zein* locus (*qγ27*) is essential to convert soft endosperm into hard endosperm in quality protein maize (QPM). This duplication is unstable and generally produces CNV at this locus. We conducted genetic experiments designed to directly measure DNA rearrangement frequencies occurring in males and females of different genetic backgrounds. The average frequency with which the duplication rearranges to single copies is $1.27 \times 10^{-3}$ and varies among different lines. A triplication of *γ27* gene was screened and showed a better potential than the duplication for the future QPM breeding. Our results highlight a novel approach to directly determine the frequency of DNA rearrangements, in this case resulting in CNV at the *qγ27* locus. Furthermore, this provides a highly effective way to test suitable parents in QPM breeding.

[1] State Key Laboratory of Crop Biology, College of Life Sciences, Shandong Agricultural University, Tai'an 271018, China. [2] National Key Laboratory of Plant Molecular Genetics, CAS Center for Excellence in Molecular Plant Sciences, Institute of Plant Physiology and Ecology, Shanghai Institutes for Biological Sciences, Chinese Academy of Sciences, Shanghai 200032, China. [3] University of the Chinese Academy of Sciences, Beijing 100049, China. [4] Department of Agronomy and Horticulture, Center for Plant Science Innovation, Beadle Center for Biotechnology, University of Nebraska, Lincoln, NE 68588-0665, USA. [5] School of Agriculture and Biology, Shanghai Jiao Tong University, Shanghai, China. [6] These authors contributed equally: Hongjun Liu, Yongcai Huang, Xiaohan Li. *email: yrwu@sibs.ac.cn

CNV is a major type of structural variation which often leads to genetic plasticity and phenotypic diversity in plants[1–5]. CNV is broadly detected in form of duplications, deletions, and insertions in the whole genome of outcrossing and self-pollinated species. Many studies have shown that CNV affects gene expression patterns in crop breeding. For instance, the *MATE1* locus confers a better tolerance to aluminum stress because increased *MATE1* copy number results in an amplification of total expression of this gene in a dosage effect. This QTL is a tandem triplication and illustrates a role of CNV in the adaption to new environments[6]. The *Grain Length on Chromosome 7* (*GL7*), carrying a tandem duplication of a 17.1-kb segment at the *GL7* locus, results in an increase in grain length and improvement of grain quality by modulating the expression of *GL7* and its nearby genes. This allelic duplication occurred before domestication and has been selected and used for the rice breeding[7]. A similar study was also found in the *SUN* locus of tomato, in which a 24.7-kb gene duplication leads to increased *SUN* expression and, as a consequence, an elongated fruit shape compared to tomato with the ancestral copy[8].

The most abundant storage proteins in maize endosperm are the zein prolamins which are classified into four subgroups (19- and 22-kDa α-zeins, 50-, 27- and 16-kDa γ-zeins, 15-kDa β-zein, 18- and 10-kDa δ-zeins)[9,10]. Zeins account for >60% endosperm proteins and are extremely deficient in the essential amino acid lysine, thereby resulting in poor nutritional value of the maize grain[11,12]. Opaque2 (O2) is a bZIP transcription factor that mainly regulates the expression of α-zein and β-zein genes[13–15]. In the *o2* mutant endosperm, the amount of zein proteins is dramatically reduced, but the total protein level remains relatively constant by a complementary increase of non-zein proteins. As a consequence, the mutant has a doubling of lysine levels[16–18]. However, the chalky and soft endosperm texture in *o2* obstructed its practical commercialization[19]. Breeding at CIMMYT (International Maize and Wheat Improvement Center) discovered genetic suppressors of the *o2* phenotype (*o2* modifiers) that could revert the soft endosperm into a vitreous and hard texture without losing the high lysine trait. The modified *o2* variety is known as QPM[20]. γ27 protein plays an important role in initiation and stabilization of RER-derived (rough endoplasmic reticulum) protein bodies (PBs), where zein proteins are synthesized and stored[21,22]. The enhanced expression of γ27 protein in QPM is essential for endosperm modification[23–28]. The major *o2* modifier (*qγ27*) results from a 15.26-kb duplication at the *γ27* locus, which increases the level of gene expression and γ27 protein abundance[29]. *qγ27* was previously designated the *Standard* allele (*S*) for bearing two copies of *γ27* gene (*Saa*, referring to the *S* allele with two identical *γ27* copies, *Sab* two copies with single nucleotide polymorphisms, SNPs)[30–32]. Since *qγ27* also exists in many accessions of teosintes examined (*Zea mays* ssp. *Parviglumis*), this duplication should occur before the maize domestication and the single-copy *γ27* alleles in modern maize might result from DNA rearrangement[29]. Indeed, early studies demonstrated that the *Sab* allele could somatically rearrange to single copies producing the recombinant alleles of *Ra* and *Rb*[30]. Although DNA rearrangement causes a reduction in γ27 expression, the absence of a visible phenotype in normal maize kernels did not allow the generation of a throughput screen of the rearranged events. In turn, this precluded the investigation of the frequency of DNA rearrangement at this locus.

The null K0326Y-Del (K-D) is a mutant QPM line generated by γ-irradiation, and entirely lacks the *γ27* locus[33]. In this study, we used the *γ27* locus as a case to examine the frequency of DNA rearrangement in different maize inbred lines. Although loss of one *γ27* copy causes no phenotype in normal inbred lines, we designed a high efficiency PCR marker to screen rearranged alleles in hemizygotes of the *S* allele and *K-D* in a large scale. We determined that the frequency with which the *S* allele rearranges to single copies, from one generation to another, is in the order of $10^{-3}$ and varies among different lines. The triplication allele of *γ27* gene resulting from DNA rearrangement was also identified in the unique population of UniformMu stocks and shown to have a superior value for the future QPM breeding compared to the *S* allele.

## Results

**Genetic design for screening rearranged alleles at the *γ27* locus.** Screening rearranged alleles at the *γ27* locus occurring at meiosis based on phenotype is impossible, because a single copy of this gene is sufficient to maintain the vitreous phenotype in normal maize kernels[29]. A reduction of γ27 affects endosperm modification in QPM, causing an opaque phenotype in the kernel. The formation of vitreous endosperm is regulated by a complex developmental process and many other factors may interfere with sorting of events that specifically result from DNA rearrangement at this locus. Thus, we resorted to the molecular screening.

The duplication of this locus contains two copies of *γ27* gene (GRMZM2G138727 or Zm00001d020592), GRMZM2G565441, GRMZM2G138976, and GRMZM5G873335 based on B73_vs3. In B73_vs4, the latter three were annotated as one gene (Zm00001d020593) with fifteen exons encoding the *ARID-transcription factor 4* (*ARID4*). The duplicated *ARID4* gene has a deletion in its 3' region, resulting in the absence of four exons. We developed a polymorphic PCR primer pair (0707-1) flanking this deletion that could successfully test whether an inbred line bears the duplication or a single copy depending on the presence of two {large (0707-1L) and small (0707-1S)} or one (0707-1L or 0707-1S) PCR bands[29]. Since the PCR banding is a dominant molecular marker, the single copy allele produced from rearrangement during gametogenesis would be masked by the other allele without rearrangement in a heterozygote for this locus. The K0326Y genome is being sequenced and assembled in our lab, yielding a long continuous sequence that was identified to cover this region. Several primer pairs were designed to amplify the junction of the deletion in K-D (Fig. 1a). The PCR products were sequenced and it was determined that the deletion spanned 1.38 Mb (Fig. 1a). The mismatched alignment of nucleotides at the junction between K-D and K0326Y probably resulted from false repair of DNA damage after the γ-irradiation (Fig. 1a). Consistent with K-D being a deletion line, the accumulation of γ27 protein is completely missing in the SDS-PAGE gel (Fig. 1b). We speculated that rearranged single-copy alleles from gametogenesis could be identified in hemizygotes when different lines with *Saa* (Mo17 and K0326Y) or *Sab* (W22 and A188) were crossed to K-D, because the PCR bands could only be amplified from the parents contributing the duplication allele. We made reciprocal crosses between K0326Y, W22, A188, and Mo17 with K-D. Over 45,000 hemizygous seeds in total were germinated and the leaf genomic DNA was extracted for PCR amplification. A representative event resulting from DNA rearrangement was shown in Fig. 1c, which only gave rise to the 0707-1L band. In total, 58 rearranged events were identified. The frequency of DNA rearrangement at this locus in different lines and sex is summarized in Table 1. The average frequency with which the duplication rearranged to single copies from one generation to another is on the order of $10^{-3}$ ($1.27 \times 10^{-3}$). The frequency varies dramatically among different lines, with the highest in A188 when used as female and lowest in W22 when used as male. For A188, the single copies were rearranged from the duplication at a frequency nearly three-fold higher in female ($2.67 \times 10^{-3}$) than male ($0.90 \times 10^{-3}$) gametogenesis. In K0326Y, W22, and

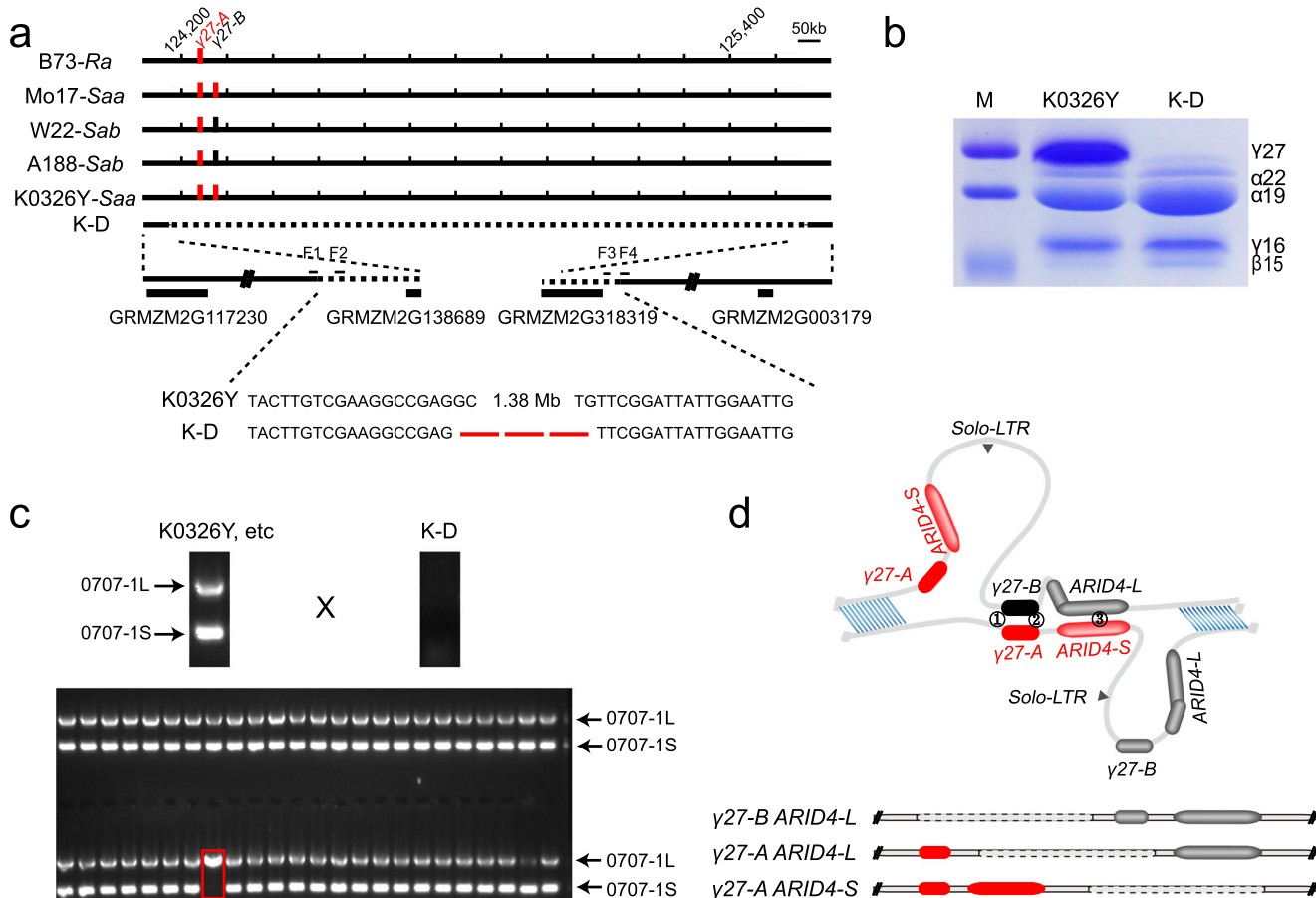

**Fig. 1 Genetic designation of measurement of DNA rearrangement frequency at the γ27 locus. a** Structure of the γ27 locus. B73 bears the *Ra* allele, Mo17 and K0326Y the *Saa* allele, and W22 and A188 the *Sab* allele. The K-D line bears a 1.38-Mb deletion at the locus, entirely removing the γ27 locus. The junction ends of the deletion are shown in an enlarged window, wherein four genes near the two ends are indicated. GRMZM2G117230 and GRMZM2G003179 are located outside of the deletion and retain in the K-D genome, whereas GRMZM2G138689 and GRMZM2G318319 reside in the deletion, thereby missing in K-D; **b** SDS-PAGE analysis of zein proteins in K0326Y and K-D. Total zein from 200 μg of corn flour was loaded in each lane. Each subgroup of zeins is indicated beside the gel. M, protein markers from top to bottom correspond to 25, 20, and 15 kDa; **c** Screening rearranged events by PCR amplification of genomic DNA from crosses of different inbred lines with K-D. A rearranged event is indicated by a red box. 0707-L (2387 bp) and 0707-S (464 bp) are the larger and smaller PCR bands amplified from *ARID4-L* and *ARID-S*, respectively; **d** Diagrammatic representation of nonallelic homologous recombination mediated DNA rearrangement at the γ27 locus for the *Sab* allele. ①, ②, and ③ indicate the locations that DNA rearrangement may occur at this locus.

**Table 1 Frequencies of DNA rearrangement at the γ27 locus in different inbred lines.**

| Hybrid combination | Individual plant count | 0707-1L | 0707-1S | Rearrangement plant count | Frequency (‰) |
|---|---|---|---|---|---|
| K0326Y × K-D | 7,650 | 14 | 3 | 17 | 2.22 |
| K-D × K0326Y | 1,034 | 2 | 0 | 2 | 1.93 |
| W22 × K-D | 16,904 | 12 | 2 | 14 | 0.83 |
| K-D × W22 | 2,887 | 2 | 0 | 2 | 0.69 |
| A188 × K-D | 2,991 | 6 | 2 | 8 | 2.67 |
| K-D × A188 | 1,114 | 1 | 0 | 1 | 0.90 |
| Mo17 × K-D | 9,819 | 8 | 2 | 10 | 1.02 |
| K-D × Mo17 | 3,334 | 2 | 2 | 4 | 1.20 |
| All | 45,733 | 47 | 11 | 58 | 1.27 |

K0326Y and Mo17 bear the *Saa* allele, and W22 and A188 bear the *Sab* allele

Mo17, the frequencies were not apparently affected by the direction of cross (Table 1).

A model for DNA rearrangement at this locus is shown in Fig. 1d. Loss of the left or right copy of γ27 gene depends on the position of the exchange relative to that of the two copies in the displaced paired segments. Among 58 events, 47 produced the

single 0707-1L (Table 1) and 11 produced the 0707-1S band, indicating most rearrangement occurs before the 0707-1S/L site. To precisely determine the rearranged sites, eight pairs of SNPs (SNP1/1* to SNP8/8*) between the duplicated fragments were identified in the four inbred lines. K0326Y contains seven SNPs (Fig. 2a), W22 and Mo17 each has six (Figs. 2b, d), and A188 only

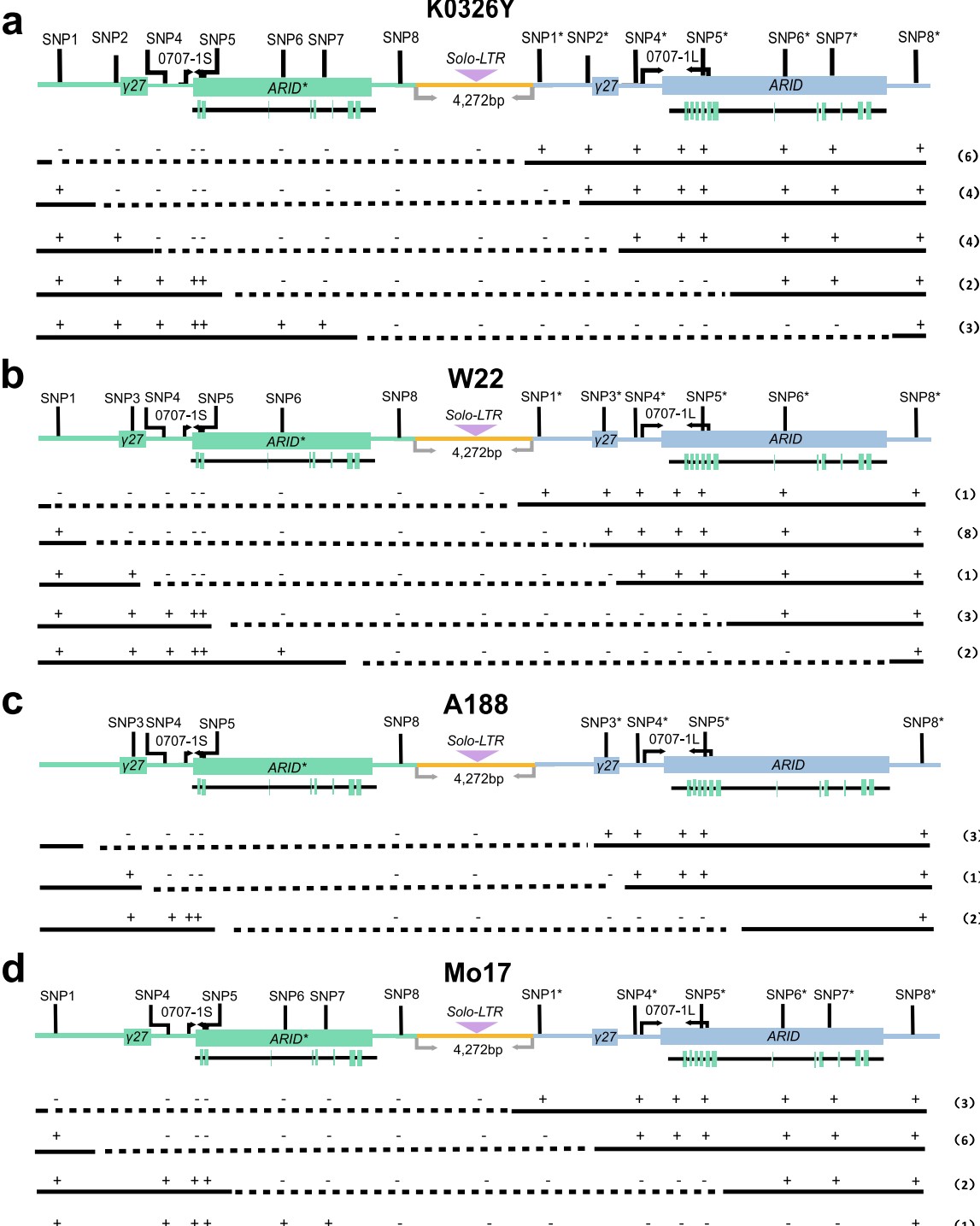

**Fig. 2 Structural analysis of DNA rearrangement at the γ27 locus. a–d** Structure of the γ27 locus duplication. Nine pairs of polymorphisms (SNP1/1*, SNP2/2*, SNP3/3*, 0707-1S/L, SNP4/4*, SNP5/5*, SNP6/6*, SNP7/7*, and SNP8/8*) for analysis of rearranged sites are indicated. Based on SNP detection, the regions wherein rearrangement occurred were drawn in dotted lines. The number of events for each region was indicated in the bracket beside. **a** K0326Y; **b** W22; **c** A188; **d** Mo17.

bears four (Fig. 2c). We successfully amplified and sequenced 52 events (19, 15, 6, and 12 for K0326Y, W22, A188, and Mo17, respectively) and summarized the results of DNA rearrangement in Fig. 2. We found that the *Solo-LTR* retrotransposon was lost in all rearranged events. Since K0326Y and Mo17 have two identical γ27-A copies, the 31 events were roughly classified into two types: 23 γ27-A ARID4-L and 8 γ27-A ARID4-S (Figs. 2a, d). Because W22 and A188 have the *Sab* allele due to a conserved

polymorphism (SNP3/3*) that distinguishes the two copies of γ27 gene, the 21 events could be categorized into three types: before SNP3/3*, between SNP3/3* and 0707-1S/L, and after the latter, which resulted in formation of three kinds of genic recombinants, γ27-B ARID4-L, γ27-A ARID4-L and γ27-A ARID4-S each with 12, 2, and 7 events, respectively (Figs. 1d, 2b, c). These results demonstrate that more than half of the events were rearranged at a site before SNP3/3*.

**Screening rearranged alleles with three copies of *γ27* gene**. DNA rearrangement of *Sab* gave rise to the single-copy allele of *Ra* or *Rb* in one chromosome, and may also simultaneously produce the three-copy allele of *Rabb* or *Raab* in the other homologous chromosome (Fig. 1d). However, because the triplication allele has an identical PCR banding phenotype as the duplication allele, it is not feasible to isolate this type of events in the populations used above by the primer pair 0707-1. The UniformMu Transposon Resource is a genetic population that carries the duplication of *γ27* in a partial W22 background and is propagated individually from single seeds[34]. Different types of alleles may exist in this unique population. The triplication allele is expected to be identified in different UniformMu stocks. 104 UniformMu stocks, which were obtained from the Maize Genetics Cooperation stock Center (MGCSC) (https://www.maizegdb.org/uniformmu) and propagated in our lab, were germinated for extraction of leaf DNA. Real-time PCR was performed to determine the copy number of *γ27* gene with the genomic DNA of each UniformMu stock. B73 and W22 known to bear a single copy (*Ra*) and the duplication (*Sab*), respectively, were used as a control, whereas the single-copy gene *prolamin-box binding factor1* (*Pbf1*) was employed as the internal control. Consistent with their copy number variation of *γ27* gene, W22 genomic DNA produced a doubling of amplicons compared to B73, confirming that this screening system is reliable. Among the 104 UniformMu stocks, five (Mu07874, Mu03708, Mu02253, Mu05783, and Mu06512) were shown to yield the amount of amplicons three-fold higher than B73 (Fig. 3a). However, only four stocks (Mu00350, Mu01653, Mu03493, and Mu06712) gave rise to levels of *γ27* amplicons similar to that

produced from the standard W22. In contrast, 95 stocks had similar levels of amplicons with B73, indicating that most UniformMu stocks contain a single-copy allele of *γ27* gene. Randomly chosen stocks with one-, two- and three-fold levels of amplicons compared to B73 were analyzed for the zein protein accumulation by SDS-PAGE. Consistently, the amount of γ27 protein is generally proportional to the DNA content of this gene, when visualized by Coomassie Brilliant Blue staining (Fig. 3b), indicating copy number variation is a critical factor determining the protein levels.

To further verify that the five stocks accumulating the highest levels of γ27 protein truly harbor the triplication allele, the coding sequences of *γ27* gene were amplified and sequenced. The ratios of sequences from A and B copies were all 1:2 ($x^2_{0.05} < 3.84$), while the ratio in the standard W22 was 1:1, confirming that these alleles are a triplication and all should be the *Rabb* type. The triplication of *γ27* gene could also be verified by the allelic expression analysis. The stock Mu06512 was used as an example for this test. Inbred XF134 bears an 18-bp deletion in the *γ27* coding sequence. Because the endosperm is triploid (two from the female and one from the male) and the standard W22 has two copies of *γ27* gene, the cDNA numbers of normal (from the standard W22) and short (from XF134) *γ27* alleles were 4:1 in the cross of W22 × XF134. Due to three copies of *γ27* gene in Mu06512, the ratios were increased to be 6:1 and 2:3 in the crosses of Mu06512 x XF134 and XF134 x Mu06512, respectively ($x^2_{0.05} < 3.84$).

Taken together, new alleles with three copies of *γ27* gene were screened from UniformMu stocks and were shown to

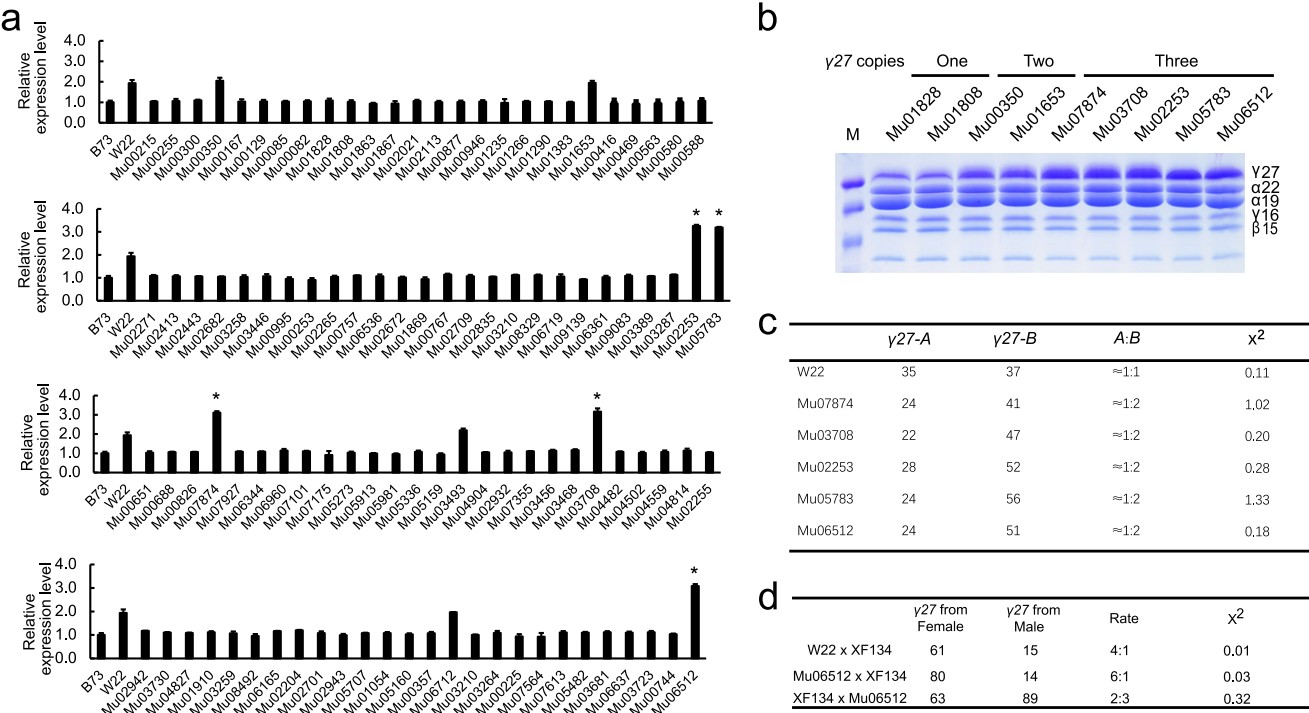

**Fig. 3 Screening rearranged alleles with a triplication of *γ27* gene in UniformMu stocks. a** Quantitative analysis of *γ27* copies in different UniformMu stocks by genomic real-time PCR. The single-copy gene *Pbf* was used as the internal control; **b** SDS-PAGE analysis of zein proteins in UniformMu stocks with one, two and three copies of *γ27* gene. The level of γ27 protein is generally proportional to the copy number. The accession number for each stock is indicated above the corresponding lane. Each subgroup of zeins is indicated beside the gel. Total zein from 200 μg of corn flour was loaded in each lane. M, protein markers from top to bottom correspond to 25, 20, and 15 kDa; **c** The ratios of the A (*γ27-A*) and B (*γ27-B*) copies of *γ27* gene in different stocks. The PCR products amplified from the leaf genomic DNA with the *γ27* gene primers and cloned for sequencing, and the genomic DNA numbers of two *γ27* alleles are expected to be 1:1 (expectation rate). $x^2_{0.05} <3.84$; **d** Allelic expression of *γ27* gene in different lines. The triplet endosperm is formed by the fusion of one sperm of the male parent with the two polar nuclei of the female parent, and W22 contains two copies of *γ27* gene while XF134 contains one copy, the cDNA numbers of two *γ27* alleles (female from W22 and male from XF134) are expected to be 4:1 (expectation rate). $x^2_{0.05} < 3.84$.

Panel c:

| | γ27-A | γ27-B | A:B | $x^2$ |
|---|---|---|---|---|
| W22 | 35 | 37 | ≈1:1 | 0.11 |
| Mu07874 | 24 | 41 | ≈1:2 | 1,02 |
| Mu03708 | 22 | 47 | ≈1:2 | 0.20 |
| Mu02253 | 28 | 52 | ≈1:2 | 0.28 |
| Mu05783 | 24 | 56 | ≈1:2 | 1.33 |
| Mu06512 | 24 | 51 | ≈1:2 | 0.18 |

Panel d:

| | γ27 from Female | γ27 from Male | Rate | $x^2$ |
|---|---|---|---|---|
| W22 x XF134 | 61 | 15 | 4:1 | 0.01 |
| Mu06512 x XF134 | 80 | 14 | 6:1 | 0.03 |
| XF134 x Mu06512 | 63 | 89 | 2:3 | 0.32 |

greatly enhance the accumulation of γ27 protein compared to the duplication alleles.

**Protein Body number in *Ra*, *Sab*, and *Rabb* endosperm cells**. In maize endosperm, zein proteins are stored in well-organized protein bodies (PBs), in which γ- and β-zeins are located in the peripheral region and α- and δ-zeins are positioned in the central area. γ27, along with the paralogous γ16 (16-kDa γ-zein) and β15 (15-kDa β-zein), is thought to play a critical role in initiation and stabilization of PB formation, whereas α-zeins are mainly responsible for PB expansion[28,35,36]. The UniformMu stocks with a different copy number of the *γ27* gene did not exhibit any phenotype in the vitreousness of kernel. This indicates that the amount of γ27 protein produced from one copy of this gene is able to initiate a sufficient number of PBs required for vitreous endosperm formation. Although the silencing or complete deletion of *γ27* gene in QPM and normal maize significantly reduced the PB number[28,33,36], it was not known how the stoichiometry of this protein affects protein body formation in a common genetic background when the *O2* gene is normal. The accumulation of γ27 protein gradually increased in the stocks with the *Ra*, *Sab*, and *Rabb* allele, respectively, which allowed us to test the above question. We examined 18-DAP (days after pollination) developing endosperm cells by transmission electron microscopy (TEM). The PB size in the three stocks appeared to be similar (Fig. 4a–c). We then statistically inspected the PB density by measuring the distance of two adjacent PBs in the fifth layer of endosperm cells. The density of PBs was apparently highest in *Rabb* and lowest in *Ra* (Fig. 4d). This observation was consistent with the primary function of γ27 being in PB initiation and stabilization rather than PB size expansion.

**Endosperm modification by the *Rabb* allele**. The duplication of *γ27* gene, and its associated transcript and protein increase, is the most significant *o2* modifier (*qγ27*). We previously sequenced the gene copies in 38 QPM lines and found that 22 carried the *Sab* allele, and 16 contained the *Saa* allele. This gene duplication is

necessary rather than sufficient for endosperm modification in QPM, which requires other modifiers to fully specify vitreous endosperm formation[37]. Since γ27 protein has a major effect on endosperm modification and the extent of modification is positively correlated to the level of γ27[23,27,33], we tested whether the triplication allele which synthesizes the highest level of γ27 might be sufficient to convert the opaque endosperm into vitreous one in the UniformMu stock background. Previous studies have demonstrated that silencing of 19- and 22-kDa α-zein expression by RNA interference (RNAi) could mimic the opaque phenotype caused by the recessive *o2* mutation[36,38]. *αRNAi* is dominant and gives rise to even a higher level of lysine compared to *o2* mutants[38]. As shown in Fig. 5a, the *αRNAi* line in the B73 background is completely opaque, due to a dramatic reduction of α-zeins. Since B73 contains the *Ra* allele, the accumulation of γ27 is markedly lower than that in Mu06512, which bears the *Rabb* allele (Fig. 5b). To compare the endosperm phenotype in the same W22 background, the *αRNAi* gene was introgressed into Mu06512 for five generations. A representative ear from a self-pollinated BC5F1 plant (*Rabb/Rabb;αRNAi/+*) is shown in Supplementary Fig. 1a. All progeny seeds are homozygous for the *Rabb* allele, three quarters of which inherited the *αRNAi* gene but did not reproduce the opaque phenotype (Supplementary Fig. 1b), although the expression of α-zeins was dramatically suppressed (Supplementary Fig. 1c). The modified kernels (*αRNAi*-M) developed a similar portion of vitreous endosperm, but could be easily distinguished from the normal *Rabb* kernels which segregated from the same ear. *αRNAi*-M kernels appear to be dark and rough on the kernel surface (Supplementary Fig. 1b). The homozygous ear for both *Rabb* and *αRNAi* was generated by self-pollinating BC5F2 plants (Fig. 5a) and all kernels were well modified but also displayed a dark and rough appearance. The accumulation of α-zeins was dramatically reduced in *αRNAi* and *αRNAi*-M seeds, but *αRNAi*-M synthesized a markedly higher level of γ27 protein (Fig. 5b). When the contents of amino acids were measured, *αRNAi* and *αRNAi*-M seeds both contained a lysine level two-fold higher than that in the normal *Rabb* stock (Supplementary Fig. 2). γ27 is rich in cysteine and acts as a reservoir for storage of this amino acid[39]. Indeed, cysteine levels

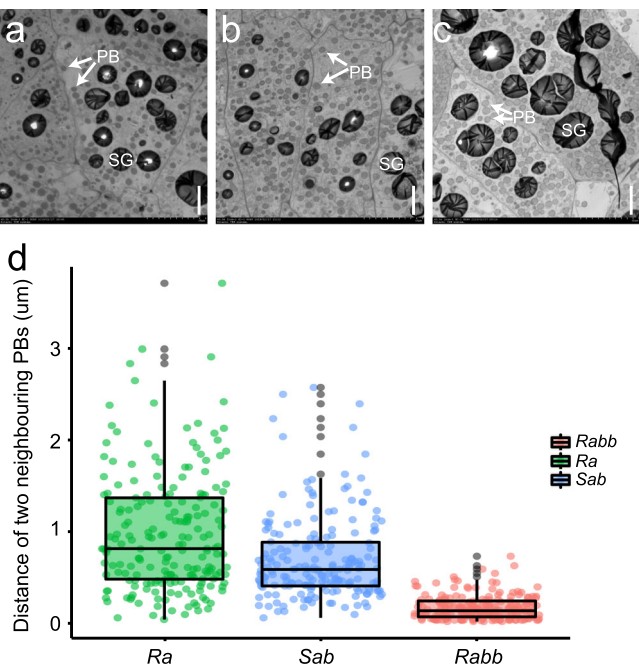

**Fig. 4 PB phenotypes in *Ra*, *Sab*, and *Rabb* endosperm cells. a** *Ra*; **b** *Sab*; **c** *Rabb*; **d** Statistical analysis of PB densities in *Ra*, *Sab* and *Rabb* endosperm cells. It is calculated by the distance of two neighboring PBs.

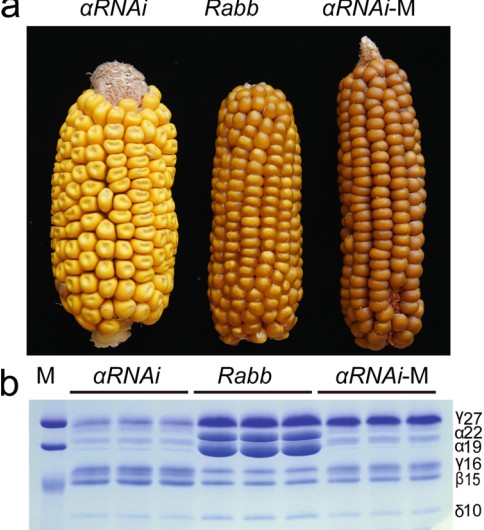

**Fig. 5 Endosperm modification in *αRNAi* by *Rabb*. a** Ear phenotypes of *αRNAi*, *Rabb* and *αRNAi*-M (modified *αRNAi* by *Rabb*); **b** SDS-PAGE analysis of zein proteins in *αRNAi*, *Rabb* and *αRNAi*-M. Three kernels for each genotype were analyzed. Each subgroup of zeins is indicated beside the gel. Total zein from 200 μg of corn flour was loaded in each lane. M, protein markers from top to bottom correspond to 25, 20, and 15 kDa.

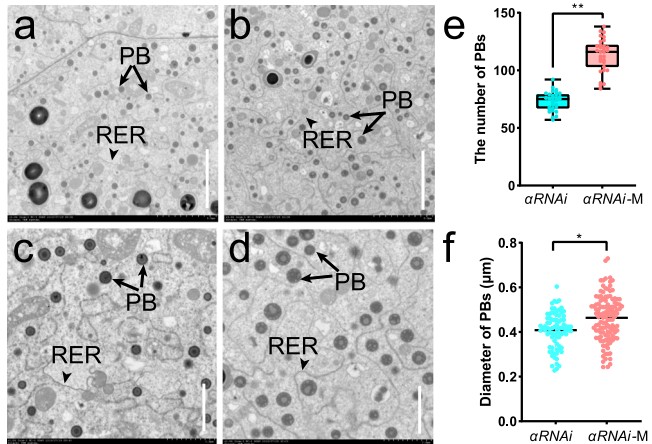

**Fig. 6 The *Rabb* allele increases the number and size of PBs in *αRNAi*-M.** **a**, **b** TEM of *αRNAi* (**a**) and *αRNAi*-M (**b**) endosperm cells at ×1000 magnification. **c**, **d** TEM of *αRNAi* (**c**) and *αRNAi*-M (**d**) endosperm cells at ×2000 magnification. SG, starch granule; PB, protein body; RER, rough endoplasmic reticulum; **e** Statistical analysis of the number of PBs in a field of view at ×1000 magnification. **\*\****P* < 0.01 as determined by Student's *t*-test; **f** Statistical analysis of PB diameters in *αRNAi* and *αRNAi*-M. **\****P* < 0.05 as determined by Student's *t*-test.

in *Rabb* and *αRNAi*-M were significantly higher than in *αRNAi*. The methionine level is regulated not only by sulfur assimilation but also the expression of 10-kDa δ-zein gene[40–42], whose encoded protein acts as a main sink of methionine in maize seed[39]. Interestingly, *αRNAi*-M seeds stored a significantly higher level of methionine than the *Rabb* stock (Supplementary Fig. 2).

To investigate the mechanism underlying endosperm modification in *αRNAi*-M, 18-DAP endosperm cells were examined. The number and size of PBs in *αRNAi*-M were both significantly increased compared to *αRNAi* (Fig. 6a–f), consistent with previous observation on endosperm modification in QPM[28,43].

## Discussion

CNVs, characterized by the addition and deletion in the number of copies of chromosomal segments or genes, have been found in individuals of different animal and plant species. In humans, 4.8–9.5% of the genome was estimated to contribute to CNVs, which play an important role in generating population diversity[44]. The maize genome exhibits vast amounts of structural variations in the form of nucleotide polymorphisms, genome size, gene content, gene density, recombination hotspots, and histone modification[4,45–47]. CNVs seem to be at least as important as SNPs in determining variations between individual maize inbred lines. By comparing the genome features of available reference data, it is apparent that the W22 genome has a higher estimated gene number (57,181) than Mo17 (38,620) and B73 (39,324). This is partly due to the number of locally duplicated genes in W22 being higher than in B73 (6034 vs. 5768)[45].

There are two main types of molecular mechanisms that lead to change in copy number, including homologous recombination (HR) and non-homologous repair mechanisms[48]. For the HR based mechanism, if a crossover occurs between the homologous sequences in the same chromosomal position, there will be no change in the structure of homologous chromosomes. However, if a crossover forms between lengths of homology in non-allelic positions on the same chromosome, this will result in gain and loss of copy number between the repeats owing to unequal crossing over (Fig. 1d). The non-homologous based mechanism uses very limited or no homology and therefore appears not to apply to the case at the *γ27* locus.

Although CNVs have a major contribution to the genome variability among maize inbred lines, the frequency of DNA rearrangement leading to CNVs is difficult to determine precisely based on phenotype, which has resulted in few cases having been reported[49,50]. Because the duplicated segments at the *γ27* locus harbor an InDel (insertion and deletion) polymorphism between the two copies of *ARID4* gene, an inbred line with or without the duplication allele could be ascertained by PCR amplification of this polymorphism[29]. Furthermore, the K-D line generated by γ-irradiation lacks the entire *γ27* locus, and consequently yields no PCR band when it is amplified with the 0707-1 primer pair (Fig. 1a–c). Thus, the rearranged single-copy alleles originating de novo could be identified in the hemizygous plants from the crosses of different inbreds (with the duplication) and the K-D line (lacking the locus). We took advantage of this genetic resolution to set up an efficient screen for the rearranged events that occurred at meiosis. We determined that the average frequency of DNA rearrangement at the *γ27* locus is $1.27 \times 10^{-3}$ (58 out of 45,733), which is 10,000-fold higher than that of spontaneous mutations in the maize genome. The spontaneous forward mutation rates of the maize *Adh1* and *C1* genes are on the order of $10^{-7}$ [51,52]. The frequencies of *Ac* mutations at the *bz-m39*(*Ac*) are $0.9 \times 10^{-3}$ on the female side and $3.7 \times 10^{-3}$ on the male size[53], close to the frequency of DNA rearrangement at the *γ27* locus in this study. We observed that the frequencies of DNA rearrangement differs in the two sexes in A188 by 3-fold (Table 1), indicating that the duplication allele rearranges more frequently in the female than in the male germline depending on the genetic background. However, using the single gametophyte sequencing, a recent study revealed that sex-specific patterns of the meiotic crossover can be accurately profiled independently of various genetic backgrounds in maize. Sex-specific patterns for meiotic crossover exhibit a higher rate (19.3 per microspore) in male than in female (12.4 per microspore)[54]. These results indicate the male has a higher recombination frequency than the female, which is opposite to our observation in A188. In addition, our data showed that the frequencies varied markedly in different inbred lines. A188 and K0326Y have the highest rearrangement frequency when used as female and male, respectively ($2.67 \times 10^{-3}$ and $2.22 \times 10^{-3}$), whereas W22 has the lowest on both sides ($0.83 \times 10^{-3}$ and $0.69 \times 10^{-3}$), indicating that more unequal crossovers are formed at this locus at meiosis in female A188 and male K0326Y than in other inbred lines.

A somatic rearrangement also seems to occur at a high frequency at the *γ27* locus in certain stocks of A188 and the resulting rearranged alleles are heritable[30]. The stability of the *γ27* locus seems to be specific to genetic stocks. It is unlikely that the *Sab-A188* allele itself is associated with its high rearrangement frequency, since the structures of all *S* alleles are similar in A188, K0326Y, Mo17, and W22 (Fig. 2). At the maize *bronze* (*bz*) locus, the frequency of homologous recombination is nearly 100-fold higher in the distal side, which is gene-dense and lacks retrotransposons, than in the proximal side, which is gene-poor and contains a large cluster of methylated retrotransposons[55]. Whether the adjacent genomic composition on either side of the *γ27* locus in A188 contributes to its rearrangement is unknown. A recent study in yeast showed that spontaneous mutation rate varies among yeast strains and is heritable. Four quantitative trait loci underlying mutation rate variation was identified[56]. Chromatin-binding factors have also been implicated in maintaining genome stability[57]. In this respect, it is intriguing to investigate whether the high frequency of DNA rearrangement is a general feature of all repeated loci or is specific to the *bz27* locus in A188.

In humans, CNV at four hot spots by non-allelic homologous recombination (NAHR) occurs at a frequency between $10^{-6}$ and

$10^{-5}$ per gamete[58]. The deletions and duplications mediated by NAHR at the α-globin locus in sperm cells also occurs at a frequency of over $10^{-5}$ [59]. The frequency of DNA rearrangement at the γ27 locus in maize is two orders of magnitude higher than that of the reported loci in humans. This observation is in agreement with the much higher genomic variation between different maize inbred lines than in individuals of the human population.

The change in copy number of a gene may change its transcript abundance, which in turn results in phenotypic variation in the population. Indeed, the copy number of γ27 gene is directly proportional to its transcript and protein levels (Fig. 3b, c)[29]. Although the duplication of γ27 gene occurred before maize domestication and was retained in the natural population, single-copy alleles can arise from mitotic and meiotic rearrangement, leaving the A or B copy at the locus[29,30] (Table 1). The instability of this duplication indicates that the change in γ27 copies seems to be neutral to maize genome evolution or adaption to the environment. Although DNA rearrangement results in an approximate two-fold reduction in the accumulation of γ27 protein, it does not confer a negative kernel phenotype in normal maize kernels. If additional copies of genes are not under positive selection pressure, they will, in theory, be rapidly purged from the population. The 19-kDa and 22-kDa α-zeins are encoded by a large number of copies, many of which are non-functional due to truncations and point mutations after the tandem duplication[60]. For the γ27 gene, only 20% inbred lines (97 out of 492) harbor the duplication[29]. This discrepancy indicates that the single-copy allele is much more stable than the duplication. The UniformMu stock population was created by introgression of a highly active MuDR (Mutator-Donald Robertson) transposon into a standard W22 inbred[34], which bears the Sab allele. We took advantage of this property to screen the triplication allele. Surprisingly, this screen resulted in identification of only four stocks (in 104) that retained the original duplication. In contrast, five contained a triplication and 95 harbored a single copy (Fig. 3a). We didn't known whether a Mu insertion near the γ27 gene would influence the DNA rearrangement frequency. It is unlikely that 95 individual seed stocks concurrently had a Mu jumping into a site near this locus. However, if the MuDR donor had a single copy of γ27 gene, at least one half of the rearranged alleles in UniformMu stocks originated from the MuDR parent in the starting population rather than from the de novo rearrangement of the W22-Sab allele later during the propagation of seed stocks.

Vitreous endosperm confers strength to withstand mechanical damage during harvesting, transportation, and storage of seed and therefore is a critical agronomic trait for maize breeding. PBs, starch granules and amorphous cytoplasmic proteins are the major factors required for vitreous endosperm formation. In the high-lysine o2 mutant, the number and size of PBs is dramatically reduced due to a greatly decreased synthesis of α-zeins, thereby resulting in the opaque endosperm phenotype. Endosperm modification in QPM involves a complex genetic action mediated by o2 modifiers, of which the duplication of the γ27 locus functions as the most major QTL. Since the enhanced expression of γ27 gene is critical for o2 endosperm modification, the structure of this duplication should be under artificial selection and stabilized in the QPM lines. In this study, we identified a triplication allele, which further elevated transcription and synthesis of γ27 protein compared to the duplication (Fig. 3b, d). The Rabb allele should be superior to Saa or Sab allele for soft endosperm modification, because it could restore kernel vitreousness in αRNAi in the UniformMu background (Fig. 5a and Supplementary Fig. 1a), due to increased number and size of PBs by this allele (Figs. 4 and 6), although it remains to be tested in other genetic backgrounds. One should keep in mind that o2 has

pleiotropic effects on endosperm development[61–63], whereas αRNAi acts to specifically reduce the expression of α-zein genes and does not affect the varied targets of O2[38]. When modifying o2 endosperm, it is possible that the other o2 modifier genes function to ameliorate those pleiotropic effects. The darker kernel phenotype may result from a higher concentration of carotenoids enriched in the modified endosperm or unknown changes in metabolic compositions accompanied by dramatic reduced α-zein and elevated γ27 protein contents. Although the lysine level in αRNAi-M seeds was slightly lower than αRNAi, probably due to the increased amount of γ27, it was still higher than 4% (Supplementary Fig. 2), reaching or exceeding the level of QPM lines[64]. Since the duplication allele is insufficient to result in a full endosperm modification, which requires a number of other modifiers, the utilization of the triplication allele will simplify and facilitate the QPM breeding, in particular when other valuable traits are introgressed into QPM hybrids.

## Methods

**Genetic materials.** The K0326Y-Deletion line (a mutant QPM line that eliminates the entire qγ27 locus, designated K-D) was generated by Dr. David Holding's laboratory at University of Nebraska, Lincoln, NE, USA. K0326Y, W22, A188 and Mo17 were detected by the 0707-1 primer pair for screening single copies of γ27 gene resulting from DNA rearrangement. The four inbred lines were reciprocally crossed with K-D, and a large number of F1 seeds were obtained for subsequent analysis of rearrangement frequency. The mature seeds were germinated in the greenhouse, and the genomic DNA was extracted from young leaves. XF134 is an inbred line that bears an 18-bp deletion in the coding sequence of γ27 gene[29]. The αRNAi line that synthesizes reduced amount of 22- and 19-kDa α-zeins was described previously[65].

**Phenotypic observation and PBs analysis.** For the cross-section observations, the mature seeds were transversely cut and placed on the platform of the stereoscope and observed for hard and soft endosperm via light microscopy. For the transmission electron microscopy, 18-DAP old endosperms with different genotypes were sliced, fixed, dehydrated, and embedded as described previously[18]. More than 20 images of transmission electron microscopy images were analyzed using ImageJ software, and all protein bodies on each slide were used for measurement and statistics. GraphPad Prism (v8.0.2) was used to calculate P-value via Student's t-tests.

**Total zein extraction.** At least 20 kernels from three individual ears were ground into fine flour using steel beads. Three biological replicates for each were performed. One hundred milligrams of flour for each sample was used for the extraction of zein proteins. Briefly, 100 mg of flour was incubated with 1 mL of zein extraction buffer (70% (vol/vol) ethanol, 2% (vol/vol) 2-mercaptoethanol, 3.75 mM sodium borate (pH 10), and 0.3% SDS) in a 2-mL on the bench for more than 2 h or overnight at room temperature. The mixture was centrifuged at 13,000 rpm for 15 min, and then 100 μL of the supernatant liquid was transferred to a new tube with an additional 10 μL 10% (wt/vol) SDS. The mixture was dried using a concentrator plus (Eppendorf) for 70 min at the condition of 45 °C and dissolved in 100 μL of distilled water. SDS-PAGE was performed to analyze the accumulation patterns of zein proteins as described previously[29]. The protein gel was visualized by staining with 0.1% Coomassie Brilliant Blue R-250 (w/v).

**Copy number and RT-qPCR analysis.** Seedling leaves of different UniformMu stocks were extracted for genomic DNA by CTAB method[66]. The DNA concentration was measured with a NANADROP 2000 instrument, and then adjusted to 10 ng/μl with water. Real-time PCR was performed with SYBR Green (TAKARA) on a Bio-Rad CFX-96 thermocycler. The fold changes in the amount of amplicons were calculated using the ΔΔCt (cycle threshold) values. The maize Pbf gene was used as an internal control. All primers used in the study are listed in Supplementary Table 1.

**Rearrangement analysis.** 45,733 seeds from reciprocal crosses of K0326Y, W22, A188, and Mo17 with K-D were screened for rearranged events at the qγ27 locus by the 0707-1 primer pair. To precisely determine the rearranged sites, seven pairs of SNPs (SNP1/1* to SNP7/7*) between the duplicated fragments were identified in K0326Y, W22, A188, and Mo17. The rearrangement site was determined by PCR amplification and sequencing of the fragment. All primers used in the study were listed in Supplementary Table 1.

**A and B copy number ratio analysis.** A and B copies of γ27 gene from W22, Mu07874, Mu03708, Mu02253, Mu05783, and Mu06512 were amplified by the

primer pair 27-F1 and 27-R1. The PCR products were recovered and cloned. More than 100 positive clones were randomly selected for sequencing. Statistical analysis was performed by calculating the number of A and B clones[29].

**Transcript ratio of different *γ27* alleles analysis.** Inbred XF134 with an 18-bp deletion in the *γ27* gene coding sequence was crossed with W22 and Mu06512. Total RNA was extracted from the 18-DAP developing endosperms with TRIzol (Invitrogen) and purified with the RNeasy Mini Kit after digestion with DNase1 (Qiagen). cDNA was synthesized with the SuperScript III First Strand Kit (Invitrogen) using 2 µg of RNA in a 20-Ml reaction system. The PCR products amplified by the primer pair 27-F1 and 27-R1 were recovered and cloned. More than 100 positive clones were randomly selected for sequencing. Statistical analysis was performed by calculating the number of clones with presence or absence of the 18-bp deletion.

**Determination of amino acid contents.** Soluble amino acid contents of kernel samples were analyzed by the Beijing Mass Spectrometry Medical Research Co. Ltd. The kernels in the middle of the mature dried ears were selected. The kernels were ground into fine powder and then further dried, and passed through a mesh sieve of 80 mesh. The measurement method was as described previously[67].

**Reporting summary.** Further information on research design is available in the Nature Research Reporting Summary linked to this article.

## Data availability

All datasets generated or analyzed during this study are included in this published article in figures (and its Supplementary Information files), or are available from the corresponding author upon request. The source data underlying Figs. 3a, 4d, 6e, 6f and Supplementary Fig. 2 are shown in Supplementary Data 1.

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

## Acknowledgements

This research was supported by the National Natural Science Foundation of China (31830063 to Y. W. and 31771799 to H.L.), the Ministry of Science and Technology of China (2016YFD0100500 to Y.W.) and Chinese Academy of Sciences (XDPB0401 to Y.W.). Research leading to the generation of the deletion line was supported a grant to D.H. from the Agriculture and Food Research Initiative competitive grant no. 2013-02278 of the USDA National Institute of Food and Agriculture.

## Author contributions

Y.W. designed experiments. H.L., Y.H., X.L., H.W., Y.D., C.K., M.S., F.L., J.W., Y.D., X.Y., X.H., X.G., L.Y., D.A., and W.W. performed experiments. H.L. drafted the paper. Y.W. and D.H. wrote the paper.

## Competing interests

The authors declare no competing interests.
