## [Peer Review File · Communications Biology]

Reviewers' comments:

Reviewer #1 (Remarks to the Author):

This manuscript by Liu et al. assesses recombination at the $\gamma 27$ locus of maize using a PCR assay of maize seedlings, quantifying the rate of the loss of the duplicated gene. Next, a few triplication alleles were identified and assessed for protein content and starch granule phenotype. These results contribute to the understanding of genome variation at the resolution of a single example locus. My major concern with this manuscript is that the biology of the locus is difficult to follow for readers not already familiar with the $\gamma 27$ locus, and there are a few inconsistencies within the paper that exacerbate the issue. For example, the three genes are listed on line 121 that do not match the genes in Figure 1a, and there are conflicting reports in the text and figures about which allele K0326Y contains. These should be corrected or clarified.

Three changes would greatly improve the readability of the paper. First, Figure 1a could be remade to better show the structure of the $\gamma 27$ locus, illustrate the variation among genotypes, and label the genes/alleles of interest. Second, this manuscript would benefit from consistently using the maize nomenclature for nonmutant alleles with genotypes listed next to alleles (i.e. Ra-B73, Saa-Mo17, etc.) as described on the Maize GDB website. Third, supplemental figures are not referenced in the text, making following the findings in the manuscript very difficult. This should be corrected.

Minor comments:

Lines 154-158 – Statements about significance should be accompanied by p-values.

Line 203 - In assessing UniformMu stocks, it is surprising that so many lines have different genotypes than W22 given the rate of rearrangement that you find. Is the $\gamma 27$ gene near the site of the Mu introgression?

Line 209 – Please describe how protein accumulation was quantified.

Line 211-224 – Please describe the statistics used in this section in the methods of this manuscript.

Line 264 – For the lines with triplicated $\gamma 27$ locus, were there any consistent genes affected by segregating Mu insertions?

Figure 5 – The figure legend should include the genotype for each ear.

Reviewer #2 (Remarks to the Author):

The manuscript entitled « High frequent DNA rearrangement at the $q\gamma 27$ locus creates a novel allele for Quality Protein Maize breeding” report the discovery of a novel allele at the $q\gamma 27$ locus. This locus is known to suppress the Opaque2 mutant by increasing zein content of maize kernel. The $q\gamma 27$ locus carries a duplication of a zein gene allowing for more RNAmessenger to be produced and more protein to be synthesized as compared to plant with single gene copy. This locus is known for long for its instability and is prone to somatic DNA rearrangements. Using quantitative PCR the authors screened a maize mutator population for a triplication event that may result in even more zein production compared to the duplication allele. The prediction that the triplication event could exist raised from the analysis of rearrangement events observed at this locus in a set of inbred line.

This work was successful at identifying the triplicated allele and to also demonstrate, as expected, its positive effect on zein protein content of kernel. This report illustrates how the careful study of structural variation can lead to predict unknown yet allele. However, such prediction can likely be done each time a gene dosage effect is observed for an allele.

This manuscript also reports data about frequency of rearrangement at this locus in a large set of crosses. The results reported however do not seem to provide major original outcomes or validate any hypothesis. It is rather descriptive. In the discussion the rearrangement frequencies observed in this work are compared with other published work such as for instance, frequency of spontaneous mutation in maize. I am not sure what is the message here. If there is no way to produce more meaningful comparisons or to provide new original clues/hypothesis about the possible mechanisms underlying rearrangement frequencies variation in assessed genotypes, I would suggest to remove this part of the manuscript or at least to extensively shorten it.

A point needs further clarification. While the duplication event was shown to be a pre-domestication event largely retained in natural populations (of teosinte ? of maize landraces ?), the authors suggest that the change in copy number is neutral for maize. But only 20% of maize inbred lines harbor the duplication and for the authors this is a consequence of the instability of this locus rather than a selection effect. I wonder if there is any lines of evidences that this locus was more stable in teosinte which is unlikely if CNV is neutral ?

As we are all well aware, there is now many high-quality maize whole genome sequence assemblies available. Some less valuable teosinte assemblies are also available. I think that the study of rearrangement event by comparing this locus in these assemblies should reinforce the results of this work by providing more precise breakpoints. By examining these breakpoints it should be possible to draw more convincing hypothesis/evidences about the mechanisms involved in these rearrangements.

This manuscript needs a lot of writing improvement. There is a lot of typos (Sab two copies with simple nucleotide polymorphisms, SNPs) and sentences with little information such as "DNA rearrangement is a major genetic driver that causes CNVs in different maize inbred lines » Can the authors cite other causes of CNVs ?

Many sentences are clumsy:

"Many studies have shown that CNV results in change in numbers of DNA segments that affect gene expression patterns in crop breeding" : CNV is a CNV !

"For instance, the MATE1 locus confers a better tolerance to aluminum stress because increased MATE1 copy number enhances the expression of this gene" Here, I think the authors have in mind an amplification of total expression (dosage effect) rather than a gene induction effect ('enhances' could be ambiguous in this context)

"In this study, we used the $\gamma 27$ locus as a model to examine the frequency of DNA rearrangement in different maize inbred lines"

This is not the purpose of this work. The $\gamma 27$ locus is the only one examined and is not taken as a model to examine the frequency of DNA rearrangement (whatever the region of the genome)of maize

"we designed a high efficient PCR marker to screen rearranged alleles in hemizygotes of the S allele and K0326Y-Del in a large scale." This allele should be described first with convenient references.

"Screening rearranged alleles at the $\gamma 27$ locus occurring at meiosis based on phenotype is impossible,

because a single copy of this gene is sufficient to maintain the vitreous phenotype in normal maize kernels²⁹ » Another reason is this locus is prone to somatic DNA rearrangement.

“Thus, we resorted to the molecular screening. » ?

Reviewer #3 (Remarks to the Author):

Maize is one of the most important crops for food, feed and fuel. More importantly for feed. However, due to deficiency of some essential amino acids, such as lysine and tryptophan, the efficiency of maize utilization is limited in feed industry. More than half a century ago, Oliver Nelson and Edwin Mertz at Purdue University found the maize $\alpha 2$ mutant containing a doubling of lysine levels and since then opened the prelude of QPM breeding, which leads to better nutritional balance of maize lines. While QPM lines are the most successful story as their both good nutritional and agronomic qualities. Through the study of the genes, proteins and biological mechanisms that changing the kernel endosperm from soft to hardness, Liu et al. indicated that the 27-kDa γ -zein gene acted as a key regulator in forming the endosperm. They found that there was a 15.26kb-duplication, which contained the 27-kDa γ -zein gene itself that strengthen the kernel hardness. So far the form mechanism of this duplication is still unknown. Therefore, this team designed a very-well genetics model and large-scale test with various copy of 27-kDa γ -zein gene. The data in the manuscript exhibited a stable rearrangement rate in various copies and genetics background. Moreover, the authors also investigated the sex differences in male and female rates that the possibility of rearrangement events may happen. In order to serve the QPM breeding, Liu et al. selected several W22 mutants that contain three copies of 27-kDa γ -zein genes to cross with a α RNAi lines to prove that the three copies lines may modify the soft endosperm directly.

Overall, this is a very-well design and classical study in genome rearrangement research. The experiments and results are well supported with the solid data. Meanwhile, there are some minor comments outlined as below:

- 1) Line 96, delete “.”;
- 2) Line 163, “0707-1S/L” to “0707-1L”;
- 3) Line 174, “before SNP2/2*, between SNP2/2* and 0707-1S/L, and after the latter,” to “before SNP2*, between SNP2 and 0707-1L, and SNP7*,”
- 4) Line 178, “before SNP2/2*” to “before SNP2*”
- 5) Line 188, “seeds” to “seed”
- 6) Line 287, “higher than that in the normal Rabb stock.” Add Fig.S2
- 7) Line 765; “K0326Y and Mo17 bear the Saa allele, and W22 and A188 bear the Sab allele” to “Mo17 bears the Saa allele, and W22, K0326Y and A188 bear the Sab allele”;
- 8) Line 910; add The representative meaning of asterisk and arrow;

Other points comments:

- 1) Table 1: Is the population sample size one of the reasons for the difference in the frequency of reciprocal cross rearrangements between parents and parents?
- 2) Why is UniformMu Stocks material selected for rearrangement experiment, what is its relationship with other inbred lines, and what is the explanatory significance of this rearrangement experiment?
- 3) Why Mu6512 was selected for the next experiment?
- 4) Why maize seeds were selected 18 days after pollination for scanning electron microscope (SEM) observation, and whether the number of protein body extracts during this period was affected by other conditions?

Reviewers' comments:

Reviewer #1 (Remarks to the Author):

This manuscript by Liu et al. assesses recombination at the $\gamma 27$ locus of maize using a PCR assay of maize seedlings, quantifying the rate of the loss of the duplicated gene. Next, a few triplication alleles were identified and assessed for protein content and starch granule phenotype. These results contribute to the understanding of genome variation at the resolution of a single example locus. My major concern with this manuscript is that the biology of the locus is difficult to follow for readers not already familiar with the $\gamma 27$ locus, and there are a few inconsistencies within the paper that exacerbate the issue. For example, the three genes are listed on line 121 that do not match the genes in Figure 1a, and there are conflicting reports in the text and figures about which allele K0326Y contains. These should be corrected or clarified.

Response: Thanks for your comments. In line 121, four genes [$\gamma 27$ gene (GRMZM2G138727 or Zm00001d020592), GRMZM2G565441, GRMZM2G138976 and GRMZM5G873335] are predicted based on the B73_vs3 in each fragment of the $\gamma 27$ duplication (15.26 kb). In B73_vs4, the latter three were annotated as one gene (Zm00001d020593) with ten exons encoding the *ARID-transcription factor 4 (ARID4)*. Figure 1a shows the large deletion in K-D, which spans 1.38 Mb, much larger than the $\gamma 27$ duplication. The four genes indicated under the enlarged window are the genes near the two junction ends. GRMZM2G117230 and GRMZM2G003179 are located outside of the deletion and retain in the K-D genome, whereas GRMZM2G138689 and GRMZM2G318319 reside in the deletion, thereby missing in K-D. We have clarified this confusion in the figure legend.

The SNP2/2* was wrongly marked in K0326Y in Fig. 2a. The two copies of $\gamma 27$ gene in K0326Y have no polymorphism in the coding sequences, so it is the Saa allele.

Three changes would greatly improve the readability of the paper. First, Figure 1a could be remade to better show the structure of the $\gamma 27$ locus, illustrate the variation among genotypes, and label the genes/alleles of interest.

Response: Thanks for the suggestion. We have clarified the genes indicated in Figure 1a in the figure legends. We have indicated the genes of our interest ($\gamma 27$ -A in red and $\gamma 27$ -B in blue) in the new Figure 1a. This locus is compared between B73, W22, Mo17, K0326Y and K-D.

Second, this manuscript would benefit from consistently using the maize nomenclature for nonmutant alleles with genotypes listed next to alleles (i.e. Ra-B73, Saa-Mo17, etc.) as described on the Maize GDB website.

Response: They have all been designated B73-Ra, Mo17-Saa, W22-Sab,

A188-*Sab* and K0326Y-*Saa* in Figure 1a.

Third, supplemental figures are not referenced in the text, making following the findings in the manuscript very difficult. This should be corrected.

Response: They have all been corrected. Figure S1 has been described in the text as follows:

- 1) A representative ear from a self-pollinated BC5F1 plant (*Rabb/Rabb;αRNAi+*) is shown in Fig. S1a. All progeny seeds are homozygous for the *Rabb* allele, three quarters of which inherited the *αRNAi* gene but did not reproduce the opaque phenotype (Fig. S1b), although the expression of α -zeins was dramatically suppressed (Fig. S1c). The modified kernels (*αRNAi-M*) developed a similar portion of vitreous endosperm, but could be easily distinguished from the normal *Rabb* kernels which segregated from the same ear. *αRNAi-M* kernels appear to be dark and rough on the kernel surface (Fig. S1b).
- 2) When the contents of amino acids were measured, *αRNAi* and *αRNAi-M* seeds both contained a lysine level two-fold higher than that in the normal *Rabb* stock (Fig. S2).

Minor comments:

Lines 154-158 – Statements about significance should be accompanied by p-values.

Response: Your suggestion is right. Statements about significance should be accompanied by p-values. Because we carried out a very large population for each cross to extract sample DNAs to calculate the frequency of DNA rearrangement, we could not do the biological replication, therefore p-values are not feasible here. To avoid this, we have corrected “significantly” into “apparently”.

In other places, p-values have been used in the figure legends.

Line 203 - In assessing UniformMu stocks, it is surprising that so many lines have different genotypes than W22 given the rate of rearrangement that you find. Is the $\gamma 27$ gene near the site of the Mu introgression?

Response: It is a very good question. We didn't know whether a *Mu* near the $\gamma 27$ gene would influence the frequency of DNA rearrangement. It is unlikely that 95 out of 104 UniformMu Stocks concurrently had a *Mu* jumping into a site near the $\gamma 27$ locus. Actually, Uniform Stocks are not in 100% W22 background (I particularly stated in the introduction that UniformMu Stocks are in a partial W22 background). The UniformMu Stock population was created by introgression of a highly active *MuDR* (Mutator-Donald Robertson) transposon into a standard W22 inbred (McCarty, 2005, Plant J.). If the *MuDR* donor had a single copy of $\gamma 27$ gene, at least one half of the rearranged alleles in UniformMu Stocks originated from the *MuDR* parent in the starting population

rather than from the de novo rearrangement of the W22-*Sab* allele later during the propagation of seed stocks. We have discussed this in the corresponding place.

Line 209 – Please describe how protein accumulation was quantified.

Response: The $\gamma 27$ protein was not quantitatively measured. It was visualized by Coomassie Brilliant Blue staining. We have rephrased this sentence as follows: Consistently, the amount of $\gamma 27$ protein is generally proportional to the DNA content of this gene, when visualized by Coomassie Brilliant Blue staining (Fig. 3b).

We have also described this in the Method: The protein gel was visualized by staining with 0.1% Coomassie Brilliant Blue R-250 (w/v).

Line 211-224 – Please describe the statistics used in this section in the methods of this manuscript.

Response: The method for statistics has been added.

Line 264 – For the lines with triplicated $\gamma 27$ locus, were there any consistent genes affected by segregating Mu insertions?

Response: For the triplicated $\gamma 27$ locus, we analyzed the ratio of cDNAs from A and B copies based on the SNPs in their coding sequences. We also analyzed the ratio of cDNAs from *Sabb* and XF134 (with an 18-bp deletion in the coding sequence). We didn't observed a phenotype segregating from Mu insertions. As you can see in Figure 5a (Middle), the ear phenotype of Mu06512 is normal. A UniformMu Stock approximately has 57 *Mu* elements and 1–2 *MuDR* copies per plant. For the lines with triplicated $\gamma 27$ locus, we didn't observe any consistent genes affected by segregating Mu insertions.

Figure 5 – The figure legend should include the genotype for each ear.

Response: The genotypes have been added.

Reviewer #2 (Remarks to the Author):

The manuscript entitled « High frequent DNA rearrangement at the q $\gamma 27$ locus creates a novel allele for Quality Protein Maize breeding” report the discovery of a novel allele at the q $\gamma 27$ locus.

This locus is known to suppress the Opaque2 mutant by increasing zein content of maize kernel. The q $\gamma 27$ locus carries a duplication of a zein gene allowing for more RNA messenger to be produced and more protein to be synthesized as compared to plant with single gene copy. This locus is known for long for its instability and is prone to somatic DNA rearrangements. Using quantitative PCR the authors screened a maize mutator population for a triplication event that may result in even more zein production compared to the

duplication allele. The prediction that the triplication event could exist raised from the analysis of rearrangement events observed at this locus in a set of inbred line.

This work was successful at identify the triplicated allele and to also demonstrate, as expected, its positive effect on zein protein content of kernel. This report illustrates how the careful study of structural variation can lead to predict unknown yet allele. However, such prediction can likely be done each time a gene dosage effect is observed for an allele.

This manuscript also report data about frequency of rearrangement at this locus in a large set of crosses. The results reported however do not seem to provide major original outcomes or validate any hypothesis. It is rather descriptive. In the discussion the rearrangement frequencies observed in this work are compared with other published work such as for instance, frequency of spontaneous mutation in maize. I am not sure what is the message here. If there is no way to produce more meaningful comparisons or to provide new original clues/hypothesis about the possible mechanisms underlying rearrangement frequencies variation in assessed genotypes, I would suggest to remove this part of the manuscript or at least to extensively shorten it.

Response: Thanks for your comments. The frequency occurring for a specific genetic event from one generation to the next is always a hot interest for a geneticist to study and indeed many kinds of genetic mutations/variations have measured for their frequency. Although we have known the instability for this locus for long, we couldn't do it until we have accumulated all these genetic materials used in this study. We compared the frequency of DNA rearrangement at this locus with frequencies of others, people will have an idea how dramatic that DNA rearrangement reshapes the maize genome, thereby causing diversity in natural maize population. Although it is currently difficult to investigate the molecular mechanism underlying this process, we have made a critical step forward. We have these genetic materials for measuring the frequency of DNA rearrangement at this locus. I believe we will know the mechanism when we isolate a mutant that affects the frequency.

A point needs further clarification. While the duplication event was shown to be a pre-domestication event largely retained in natural populations (of teosinte ? of maize landraces ?), the authors suggest that the change in copy number is neutral for maize. But only 20% of maize inbred lines harbor the duplication and for the authors this is a consequence of the instability of this locus rather than a selection effect. I wonder if there is any lines of evidences that this locus was more stable in teosinte which is unlikely if CNV is neutral ?

Response: We suggest that the change in copy number of $\gamma 27$ gene in maize is neutral for adaption. The instability of the duplication leads to a lower

percentage in maize inbred lines. We have investigated eight teosinte lines (*Z. mays* ssp. *Parviglumis*) and found six contained the duplication (Figure S8 in Liu, et al., PNAS, 2016). However, the number of teosinte lines was too small to conclude that this locus was more stable in teosinte. It is interesting to do this by crossing a *Z. mays* ssp. *Parviglumis* line with K-D to measure the frequency in the future.

As we are all well aware, there is now many high-quality maize whole genome sequence assemblies available. Some less valuable teosinte assemblies are also available. I think that the study of rearrangement event by comparing this locus in these assemblies should reinforce the results of this work by providing more precise breakpoints. By examining these breakpoints it should be possible to draw more convincing hypothesis/evidences about the mechanisms involved in these rearrangements.

Response: Yes, we have done that as shown in Fig. 2. We have sequenced all rearranged alleles. Due to lack of sufficient SNPs between the fragments of this duplication and no footprints left after rearrangement, the breakpoints could not be precisely determined at a base-pair resolution for each event. But you can see the site wherein each rearrangement occurred was already located into a small region.

This manuscript needs a lot of writing improvement. There is a lot of typos (Sab two copies with simple nucleotide polymorphisms, SNPs) and sentences with little information such as “DNA rearrangement is a major genetic driver that causes CNVs in different maize inbred lines » Can the authors cite other causes of CNVs ?

Response: The language has been polished. The “simple” has been corrected to “single”. We have deleted the sentence “DNA rearrangement is a major genetic driver that causes CNVs in different maize inbred lines”.

Many sentences are clumsy:

“Many studies have shown that CNV results in change in numbers of DNA segments that affect gene expression patterns in crop breeding” : CNV is a CNV !

Response: It has been corrected as follows:

Many studies have shown that CNV affects gene expression patterns in crop breeding.

“For instance, the MATE1 locus confers a better tolerance to aluminum stress because increased MATE1 copy number enhances the expression of this gene” Here, I think the authors have in mind an amplification of total expression (dosage effect) rather than a gene induction effect (‘enhances’ could be

ambiguous in this context)

Response: We have rephrased this as follows.

For instance, the *MATE1* locus confers a better tolerance to aluminum stress because increased *MATE1* copy number results in an amplification of total expression of this gene in a dosage effect.

“In this study, we used the $\gamma 27$ locus as a model to examine the frequency of DNA rearrangement in different maize inbred lines”

This is not the purpose of this work. The $\gamma 27$ locus is the only one examined and is not taken as a model to examine the frequency of DNA rearrangement (whatever the region of the genome)of maize

Response: We have replaced “model” with “case”.

“we designed a high efficient PCR marker to screen rearranged alleles in hemizygotes of the S allele and K0326Y-Del in a large scale.” This allele should be described first with convenient references.

Response: It has been rephrased as follows:

“The null K0326Y-Del (K-D) is a mutant QPM line generated by γ -irradiation, and entirely lacks the $\gamma 27$ locus³³. In this study, we used the $\gamma 27$ locus as a case to examine the frequency of DNA rearrangement in different maize inbred lines. Although loss of one $\gamma 27$ copy causes no phenotype in normal inbred lines, we designed a high efficiency PCR marker to screen rearranged alleles in hemizygotes of the S allele and K-D in a large scale.”

“Screening rearranged alleles at the $\gamma 27$ locus occurring at meiosis based on phenotype is impossible, because a single copy of this gene is sufficient to maintain the vitreous phenotype in normal maize kernels²⁹ » Another reason is this locus is prone to somatic DNA rearrangement. “Thus, we resorted to the molecular screening. » ?

Response: Partially modified kernels are occasionally found in QPM ears. Somatic DNA rearrangement should be a reason.

Reviewer #3 (Remarks to the Author):

Maize is one of the most important crops for food, feed and fuel. More importantly for feed. However, due to deficiency of some essential amino acids, such as lysine and tryptophan, the efficiency of maize utilization is limited in feed industry. More than half a century ago, Oliver Nelson and Edwin Mertz at Purdue University found the maize o2 mutant containing a doubling of lysine levels and since then opened the prelude of QPM breeding, which leads to better nutritional balance of maize lines. While QPM lines are the most successful story as their both good nutritional and agronomic qualities. Through the study of the genes, proteins and biological mechanisms that

changing the kernel endosperm from soft to hardness, Liu et al. indicated that the 27-kDa γ -zein gene acted as a key regulator in forming the endosperm. They found that there was a 15.26kb-duplication, which contained the 27-kDa γ -zein gene itself that strengthen the kernel hardness. So far the form mechanism of this duplication is still unknown. Therefore, this team designed a very-well genetics model and large-scale test with various copy of 27-kDa γ -zein gene. The data in the manuscript exhibited a stable rearrangement rate in various copies and genetics background. Moreover, the authors also investigated the sex differences in male and female rates that the possibility of rearrangement events may happen. In order to serve the QPM breeding, Liu et al. selected several W22 mutants that contain three copies of 27-kDa γ -zein genes to cross with a α RNAi lines to prove that the three copies lines may modify the soft endosperm directly.

Overall, this is a very-well design and classical study in genome rearrangement research. The experiments and results are well supported with the solid data. Meanwhile, there are some minor comments outlined as below:

1) Line 96, delete “.”;

Response: Done.

2) Line 163, “0707-1S/L” to “0707-1L”;

Response: 0707-1S/L is correct. The two homologies are paired (0707-1S/L).

3) Line174, “before SNP2/2*, between SNP2/2* and 0707-1S/L, and after the latter,” to “before SNP2*, between SNP2 and 0707-1L, and SNP7*,”

Response: They are correct. These homologies are paired.

4) Line178, “before SNP2/2*” to “before SNP2*”

Response: It is correct.

5) Line 188, “seeds” to “seed”

Response: It should be single seeds. It means many different seeds.

6) Line 287, “higher than that in the normal Rabb stock.” Add Fig.S2

Response: Added.

7) Line765; “K0326Y and Mo17 bear the Saa allele, and W22 and A188 bear the Sab allele” to “Mo17 bears the Saa allele, and W22, K0326Y and A188 bear the Sab allele”;

Response: It our mistake. The SNP2/2* was wrongly marked in K0326Y in Fig. 2a. The two copies of γ 27 gene in K0326Y have no polymorphism in the coding sequences, so it is the Saa allele. It has been corrected in Fig. 2 and in the text.

8) Line910; add The representative meaning of asterisk and arrow;

Response: Arrows and asterisks indicate normal (*Rabb*) and modified (*αRNAi-M*) kernels. It has been added in the figure legend.

Other points comments:

1) Table1: Is the population sample size one of the reasons for the difference in the frequency of reciprocal cross rearrangements between parents and parents?

Response: The population is larger, the frequency is more accurate. In this case, it is unlikely that the population sample size affected the frequency difference, because each sample size is big enough. For W22 × K-D (16,904), Mo17 × K-D (9,818) and K0326Y × K-D (7,650), the frequencies didn't change when their population sizes were more than 2000. They had the largest sample number, because they produced the most amounts of seeds and we germinated them all for testing.

2) Why is UniformMu Stocks material selected for rearrangement experiment, what is its relationship with other inbred lines, and what is the explanatory significance of this rearrangement experiment?

Response: UniformMu has a large number of Stocks in a relatively same genetic background. They bear the duplication, which may produce the triplication by DNA rearrangement. Other inbred lines with the duplication don't have enough stocks for screening. The triplication is predicted for existence in the UniformMu Stocks by studying the mode of DNA rearrangement and has better potential for QPM breeding.

3) Why Mu6512 was selected for the next experiment?

Response: Since the *Sabb* allele in Mu6512 is the same with other triplications, it was randomly chosen for no particular reason. We could not do all backcrossing into the five stocks at the same time.

4) Why maize seeds were selected 18 days after pollination for scanning electron microscope (sem) observation, and whether the number of protein body extracts during this period was affected by other conditions?

Response: 18-DAP represents a time point for high efficient endosperm filling. The starch and proteins are actively synthesized. The sample is easy to process.

REVIEWERS' COMMENTS:

Reviewer #1 (Remarks to the Author):

The authors have appropriately addressed my comments.

Reviewer #3 (Remarks to the Author):

The new version of this manuscript has been improved. All my concerns for this manuscript has been addressed. I recommend this manuscript to be published as the current version.